# VAX014, an Oncolytic Therapy, Reduces Adenomas and Modifies Colon Microenvironment in Mouse Model of CRC

**DOI:** 10.3390/ijms24129993

**Published:** 2023-06-10

**Authors:** Shea F. Grenier, Mohammad W. Khan, Katherine A. Reil, Savannah Sawaged, Shingo Tsuji, Matthew J. Giacalone, Mengxi Tian, Kathleen L. McGuire

**Affiliations:** 1Department of Biology, Molecular Biology Institute, San Diego State University, San Diego, CA 92182, USA; s.grenierdavis@gmail.com (S.F.G.); drmwasim@gmail.com (M.W.K.); shsawaged@gmail.com (S.S.); mengxi0607@gmail.com (M.T.); 2Vaxiion Therapeutics, San Diego, CA 92121, USA; kreil@vaxiion.com (K.A.R.); shingotsuji@vaxiion.com (S.T.); mjgiacalone@vaxiion.com (M.J.G.)

**Keywords:** colorectal cancer, immunotherapy, anti-cancer immunity, probiotics, gut microbiota

## Abstract

Colorectal cancer (CRC) remains the third most common form of cancer and, despite its reduced mortality, results in over 50,000 deaths annually, highlighting the need for novel therapeutic approaches. VAX014 is a novel clinical-stage, oncolytic bacterial minicell-based therapy shown to elicit protective antitumor immune responses in cancer, but it has not been fully evaluated in CRC. Here, VAX014 was demonstrated to induce oncolysis in CRC cell lines in vitro and was evaluated in vivo, both as a prophylactic (before spontaneous development of adenomatous polyps) and as a neoadjuvant treatment using the Fabp-CreXApc^fl468^ preclinical animal model of colon cancer. As a prophylactic, VAX014 significantly reduced the size and number of adenomas without inducing long term changes in the gene expression of inflammatory, T helper 1 antitumor, and immunosuppression markers. In the presence of adenomas, a neoadjuvant VAX014 treatment reduced the number of tumors, induced the gene expression of antitumor T_H_1 immune markers in adenomas, and promoted the expansion of the probiotic bacterium *Akkermansia muciniphila*. The neoadjuvant VAX014 treatment was associated with decreased Ki67 proliferation in vivo, suggesting that VAX014 inhibits adenoma development through both oncolytic and immunotherapeutic effects. Combined, these data support the potential of VAX014 treatment in CRC and “at risk” polyp-bearing or early adenocarcinoma populations.

## 1. Introduction

Colorectal cancer (CRC) is increasing in incidence in adults less than 55 years of age and will result in ~50,000 deaths in the U.S. this year [1], underscoring the need for effective therapeutic approaches. The risk factors for CRC include familial history, smoking, obesity, and inflammatory bowel diseases [2]. A complex and multifaceted disease, the progression and clinical outcome of CRC are affected by the immune system and microorganism populations colonizing the intestinal environment (i.e., the gut microbiota). Current neoadjuvant therapies, such as immunotherapies, chemotherapeutic agents, and targeted molecular therapies, induce responses in specific populations of CRC patients and are affected by drug resistance mechanisms [3]. Given these limitations, novel therapeutic strategies targeting the CRC microenvironment, including adenomas, specific microbiota taxa, and immune responses, may prove more effective in treating or reducing the incidence of CRC.

Approximately 95% of CRC cases develop from non-cancerous adenomatous polyps, or adenomas, that transform into malignant adenocarcinomas through the accumulation of histological and genomic alterations in a series of well-defined steps [1,4]. Adenoma initiation through common mutations of the adenomatous polyposis (Apc) gene has been replicated in genetically engineered mouse models [5]. Populations of innate and adaptive immune cells within, and on the periphery of, CRC tumors affect progression, responses to therapies, and overall survival [6]. In CRC, chronic inflammation, found in conditions such as ulcerative colitis or Crohn’s disease, can promote tumor progression and is associated with poor prognosis [7]. Alternatively, antitumor responses capable of immune-mediated tumor control, with a concomitant decrease in nonspecific inflammation, have been associated with improved clinical outcomes [8]. Antitumor immune responses in CRC are most associated with a T_H_1 CD4^+^ T cell response, which promotes the proliferation and survival of effector, CD8^+^ cytotoxic T lymphocytes. The expression of T_H_1 and CD8^+^ markers, including CD8, gamma interferon (IFN-γ), T-box transcription factor 21 (T-bet), and perforin (PRF), is positively correlated with disease-free survival, reduced rates of relapse, and is a better predictor of survival than the traditional TNM staging system [9].

Reciprocal interactions with the gut microbiota can also influence immune responses to CRC and effect pathogenesis [10]. In CRC, the gut microbiota exists in dysbiosis, or microbial imbalance, identified through changes in population composition, including increased opportunistic pathogen abundance and commensal bacterium loss [11]. Altered metabolite production and increased toxicity by dysbiotic colon microbiota are thought to induce cellular injury and chronic inflammation, promoting pathogenesis and tumorigenesis [12]. Therapeutic intervention by the manipulation of CRC microbiota by probiotics, or homeostatic commensal bacteria, has shown promise in CRC [13], indicating the potential validity of this investigational approach.

VAX014 [14,15] is a clinical-stage, oncolytic-recombinant bacterial minicell (rBMC) therapy derived from *Escherichia coli* cells containing immune-attenuated lipopolysaccharide (LPS). Lacking a chromosome, VAX014 is not replication competent; however, it does contain bacterial components recognized by the immune system [16]. VAX014 is engineered to induce oncolysis through delivery of the pore-forming cytolysin perfringolysin O (PFO), inducing the formation of irreversible membrane pores [17]. The tumor specificity of VAX014 is mediated via Invasin, which naturally targets unligated alpha3beta1 (α3β1) and alpha5beta1 (α5β1) integrin heterodimers on tumor cells. In preclinical models of non-muscle-invasive bladder cancer, VAX014 improved survival and reduced tumor growth [14,18]. In these models, the antitumor activity of VAX014 resulted from both direct oncolysis and the ensuing induction of protective CD4^+^ and CD8^+^ antitumor immune responses [19]. These therapeutic actions suggest potential relevancy in other immune-associated cancers, where direct VAX014 interaction with tumors is feasible.

This study evaluates VAX014 in the context of CRC development for the first time using in vitro methods and a preclinical animal model. The data presented here suggest that VAX014 is capable of inducing oncolysis through PFO toxin delivery in vitro, and reducing adenoma development in vivo, through the modification of the colonic environment. Importantly, VAX014 was effective in reducing polyp abundance both prior to and in the presence of macroscopic tumor development, using both prophylactic and neoadjuvant delivery, respectively. Ki67 expression levels showed that proliferation was reduced in VAX014-treated polys. While long term immune changes were not observed in any circumstances, tissues adjacent to large adenomas showed evidence of T_H_1-mediated immunity after neoadjuvant therapy. Combined, this preliminary evidence supports the fact that VAX014 warrants further investigation as a potential treatment for “at risk” populations of polyp-bearing or early-stage adenocarcinoma patients or as a neoadjuvant treatment in CRC.

## 2. Results

### 2.1. VAX014 Kills Colon Adenocarcinoma Cells In Vitro

The expression of target α3β1 and/or α5β1 integrins, required for VAX014 mediated binding and internalization [14], was validated in three human and murine colon adenocarcinoma cell lines (Figure 1A–C). VAX014 treatment in the same CRC cells exhibited dose–response, PFO-dependent, cytotoxicity after 2 h (Figure 1D) and 20 h (Appendix A) coincubations, whereas VAX-I, a control rBMC expressing Invasin but lacking the PFO payload, had no effect. Dose-dependent VAX014 induced oncolysis in CRC and was verified with lactate dehydrogenase release (Figure 1E). Target cell membrane permeabilization was observed by propidium iodide staining (Figure 1F) and DAPI nuclear staining (Appendix A) in cells treated with VAX014, but not with VAX-I or saline, indicating that the delivery of PFO is required for cell lysis. Together, these assays validate the targeting and oncolysis of CRC via membrane permeabilization by VAX014 treatment.

### 2.2. Prophylactic VAX014 Treatment Reduces Tumor Load In Vivo

The therapeutic efficacy of VAX014 in vivo was evaluated in Fabp-CreXApc^fl468^ using weekly intrarectal dosing and saline treated animals as controls. VAX-I was not included in in vivo adenoma studies since treatment was previously shown not to induce antitumor activity in bladder cancer models [19] and coincubation did not affect CRC cell viability (Figure 1). Naïve Fabp-CreXApc^fl468^ were observed to spontaneously develop adenomas at or around 14 weeks of age and to survive from 26 to 32 weeks. Adenomas developed over time (Appendix A; *p* = 0.04) and ranged from 0.5 to 9 mm^2^ in size, with large tumors, ≥4 mm^2^, developing less frequently. Prophylactic VAX014 treatment in animals 8–13 weeks of age, with 6 total doses, significantly reduced polyp development at both 14 weeks (Figure 2A; *p* = 0.05) and 26 weeks of age (Figure 2B; *p* = 0.05). In addition, large tumor frequency (≥4 mm^2^) at 26 weeks of age was also significantly reduced by prophylactic VAX014 treatment (Figure 2C; *p* = 0.05), suggesting long-term inhibition of adenoma development. Together, these data indicate that prophylactic VAX014 treatment, before macroscopic adenoma development, has lasting inhibitory effects on polyp development in the Fabp-CreXApc^fl468^ model of colon cancer.

### 2.3. Neoadjuvant VAX014 Treatment Reduces Adenoma Number and Tumor Cell Proliferation In Vivo

VAX014 treatment in animals 14–19 weeks of age, in the presence of detectable adenomas, significantly reduced polyp number (Figure 3A; *p* = 0.03), but not large tumor frequency (≥4 mm^2^), at 26 weeks of age (Appendix A; *p* = 0.64). The effects of VAX014 on adenoma tissues were investigated using a cohort of Fabp-CreXApc^fl468^ mice with well-developed adenomas treated at 22–24 weeks of age and sacrificed one-hour after the final treatment to allow rBMC interaction with the colonic mucosa. In this group, adenomas from VAX014-treated mice exhibited a significant reduction in Ki67 (Figure 3B, *p* = 0.02), a marker of cell proliferation, in IHC analyses. These data strongly suggest that neoadjuvant VAX014 treatment decreases tumor abundance through reduced tumor cell proliferation in vivo. Interestingly, α3 integrin (Figure 3C, *p* = 0.02) expression was also significantly decreased in these VAX014-treated tumors, but beta-1 (β1) integrin expression was not affected in this group (Appendix A, *p* = 0.25).

### 2.4. VAX014 Increases Markers of Cell-Mediated Immunity in Treated Polyps

The potential role of cellular immune activation after treatment with VAX014 was evaluated by gene expression in Fabp-CreXApc^fl468^ mice, using a panel of immune markers associated with the T_H_1 cell type immune response (CD8, Ifnγ, Prf, T-bet), inflammation (Interleukin 17 (Il-17), Interleukin 6 (Il-6)), and immune suppression, including that from both T_regs_ and MDSCs (Interleukin 10 (Il-10), Forkhead box P3 (FoxP3), Arginase (Arg-1), and Indoleamine 2, 3-dioxygenase (Ido1)). This panel was tested in three regions of colon tissues: (i) normal distal colon regions separated from areas of adenoma development, “macroscopically normal”; (ii) normal tissues directly adjacent to tumor tissues, “tumor adjacent”; and (iii) large adenomas ≥ 4 mm^2^.

The neoadjuvant treatment of VAX014 in Fabp-CreXApc^fl468^, from 14 to 19 weeks, upregulated the expression of Il-6 (*p* = 0.04) in macroscopically normal colon tissues (Figure 4A) and T_H_1 and cytotoxic cell markers including Ifny (*p* = 0.05), Prf (*p* = 0.02), T-bet (*p* = 0.02), and the immunoregulatory cytokine, Il-10 (*p* = 0.05), in the large (≥4 mm^2^) adenomas of 26-week-old mice (Figure 4B). No immune gene expression differences were observed in the tumor-adjacent tissues of neoadjuvant models (Appendix A). Importantly, the VAX014 treatment (14–19 weeks) had no effect on the expression of immune genes in the healthy distal colon tissues of treated 26-week-old Apc^fl468^ parental controls that did not develop adenomas (Appendix A), indicating that the presence of adenomas was required for VAX014 immunomodulatory activity. Prophylactic VAX014 treatment, from 8–13 weeks, also had no effect on Immune marker expression (Appendix A) despite reduced tumor burden at 26 weeks of age, suggesting that immune gene expression might return to naïve levels after treatment, or that immune changes are not observed in these animals due to low (or no) tumor burden at the time of treatment. The gene markers associated with chronic or protumorigenic inflammation, including Il-17, FoxP3, Arg-1, and Ido-1, were not altered by VAX014 in any treatment groups (Figure 4 and Appendix A). Combined, these data suggest that VAX014 treatment may promote antitumor immune responses in adenomas, as previously reported [19,20], but that it does not alter the immune microenvironment of normal tissues or induce long term immune changes in the colon.

### 2.5. VAX014 Modulates the Tissue-Associated Colonic Microbiota

The potential for neoadjuvant VAX014 to induce colonic microbiota changes was investigated, given the observed change in the immune gene expression and the adenoma reduction in Fabp-CreXApc^fl468^ mice. Colonic microbiota have been demonstrated to be important in colonic health and to be influenced by the development of adenomas/adenocarcinomas and local inflammation [10]. Tissue-associated microbiota were isolated from tumor-adjacent tissues in a subpopulation of the 26-week-old neoadjuvant mice used in the immune expression analysis, treated with VAX014 from 14 to 19 weeks. Since VAX014 rBMCs lack parental chromosomes [20], genomic contamination was not observed in the microbiota. Shotgun metagenomic sequencing produced an average of 1,008,072 high-quality paired-end reads, assembled into 15,742 contigs, per sample.

The VAX014 treatment did not significantly impact species-level alpha diversity, as measured by Shannon index (Appendix A; *p* = 0.78), or Principal Coordinate Analysis derived from weighted UniFrac beta diversity distances (Appendix A; *p* = 0.22). Taxonomic microbiome analysis revealed a significant expansion of the bacterial phylum Verrucomicrobia in VAX014-treated mice (*p* = 0.03), with a mean relative abundance of 17.1% and >1% abundance in control microbiomes (Figure 5A, Appendix A). A single species, *Akkermansia muciniphila*, is solely responsible for this increase in relative abundance in VAX014-treated microbiomes (Figure 5B). *A. muciniphila* is a mucin-degrading bacterium commonly found in the human gut and is considered to be probiotic in the microbiota [21]. Several microbiome genera were also trending towards significance, including increases in *Psuedomonas* and *Streptococcus* and a decrease in *Bacterioides*, but were not significant at the given sample size. Analyses of gene function revealed significant increases (*p* = 0.03) in pathways associated with Nucleotides and Nucleosides (*p* = 0.03) at subsystems level 1 (Figure 5C) and in category genes associated with Pyrimidines (*p* = 0.03) at subsystems level 2 (Figure 5D) in VAX014-treated microbiomes. DNA repair, a subsystems level 2 gene category, was significantly increased (*p* = 0.03) by VAX014, while the level 1 parent group, DNA metabolism, was found to be trending toward a significant increase (*p* = 0.07) at the given sample size (Figure 5D, Appendix A). This evidence suggests that VAX014 treatment alters colon microbiota populations and gene function to support the growth of probiotic bacteria.

## 3. Discussion

Here, we evaluated the therapeutic potential of VAX014, a novel rBMC-based tumor-targeted oncolytic agent, in the context of colonic adenomas and CRC. VAX014 exhibits dual therapeutic functions as an oncolytic and immunotherapeutic agent in preclinical studies [19]. The oncolytic activity of VAX014, via delivery of the PFO toxin through integrin binding [14], was initially validated in vitro using human and murine colon cancer cell lines. In this preclinical model of autochthonous colorectal adenoma, the prophylactic VAX014 treatment reduced both tumor number and the incidence of large adenomas in mice treated prior to tumor development (8–13 weeks), indicating that VAX014 may be a relevant prophylactic treatment for predisposed CRC populations [22]. In the neoadjuvant setting, VAX014 treatment (14–19 weeks) decreased adenoma number through reduced proliferation, as measured by Ki67 staining (Figure 3B). A correlated decrease in α3 integrin expression (Figure 3C) suggests that neoadjuvant VAX014 treatment results in tumors that have lowered expression of this integrin, perhaps either by oncolysis of, or selection against, tumors that are strongly positive for this marker. Additionally, a lack of β1 integrin reduction by neoadjuvant VAX014 may be due to the varying levels of expression on adenomas observed in control animals, or the multitude of α subunits to which it can dimerize, but which are not targeted by VAX014 [23]; in either scenario, further investigation is required.

Neoadjuvant VAX014 treatment also promoted the expression of antitumor immune cytokines in large adenomas. Importantly, gene markers of inflammation were not altered by VAX014, indicating that treatment is not pro-inflammatory in the colon microenvironment. The antitumor immune responses observed in VAX014-treated tumors included the upregulation of markers associated with T_H_1 type T cell immune responses, which have been demonstrated to correlate with improved prognosis in patients with colon cancer [24,25]. These findings are consistent with a previous report where T_H_1 type CD4^+^ T cells and activated CD8^+^ cytotoxic T cells mediated the clearance of orthotopic bladder cancer mouse models after treatment with VAX014 [19]. No gene expression changes in immune markers by VAX014 were observed in Apc-flox parental mice (control lacking polyp development), indicating tumor specificity. Additionally, immune gene expression was not altered by prophylactic VAX014 despite a reduction in tumor load, suggesting that immune homeostasis is not altered long-term by treatment. While a complete analysis of the immune responses involved in the polyps of VAX014-treated mice was not a goal of these studies, the evaluation of immune cell populations in VAX014-associated CRC in future studies would complement the global immune gene changes described here.

The observed increases of Il-10 expression in large adenomas with T_H_1 responses suggest that VAX014 does not overtly disturb the normal immune homeostasis required to maintain colon health. Unlike clinical settings, Il-10 appears to induce a protective effect against adenoma development in similar APC-flox animal models through the regulation of the microbiota by T cell populations [26]. Additionally, increased Il-6 gene expression was found in the macroscopically normal colonic regions of mice with adenomas. Despite being engineered to minimize proinflammatory effects, VAX014 still elicits dampened IL-6 responses in other systems [14]. Interestingly, increased Il-6 expression was limited to “macroscopically normal” tissue and was not observed in the adenomas, parental colonic mucosa, or colonic mucosa of prophylactic-treated animals. This suggests that elevated Il-6 is likely associated with essential colonic processes: epithelial proliferation [27], infection prevention [28], and wound healing [29] following resolution of colonic adenomas in response VAX014 treatment.

VAX014 also altered the gut microbiota in this autochthonous adenoma mouse model, identified through the significant expansion of the probiotic bacteria, *A. muciniphilia*, and enrichment of gene pathways: Nucleotides and Nucleosides, Pyrimidines, and DNA repair. A supplementation of *A. muciniphilia* by fecal transplantation has been shown to suppress the development of ovarian cancer in mouse models through the cytokine stimulation of antitumor CD8^+^ T cells [30]. *A. muciniphilia* has also been inversely correlated with several intestinal diseases [21], though some evidence may contradict this observation [31,32]. The relevancy of functional microbiome changes is less clear, though evidence has suggested that increased DNA repair gene abundance may be associated with homeostasis [33]. Importantly, microbiota in this study were isolated from colon tissues directly adjacent to adenomas, exhibiting the immunomodulatory and oncolytic effects of VAX014 treatment. Together, this evidence suggests that VAX014 treatment is cytotoxic to adenomas and correlates to probiotic and immunomodulatory effects that are relevant as both a prophylactic and neoadjuvant treatment. Future studies will need to fully evaluate the extent of VAX014-induced immunotherapeutic effects in CRC, and the therapeutic potential of VAX014 in related conditions, such as inflammatory diseases, which have relevant clinical applications.

## 4. Materials and Methods

### 4.1. VAX014 Isolation and Purification

VAX014, (PFO+, Invasin+), and VAX-I, (PFO−, Invasin+) rBMCs, were manufactured by Vaxiion Therapeutics, San Diego, Ca, USA as previously described [16,18,20].

### 4.2. Cell Lines

Colon cancer cell lines were purchased from American Tissue Type collection and maintained in 37 °C with 5% CO_2_. CT26.WT and SW620 were grown in RPMI1640 with L-glutamine, 10% heat-inactivated fetal bovine serum (FBS), and 1% penicillin/streptomycin (P/S). HCT116 was grown in McCoy’s 5a with 10% heat-inactivated FBS and 1% P/S.

### 4.3. Cell Viability and Lactate Dehydrogenase (LDH) Release Assays

Cells were seeded at 10,000 cells per well and grown to confluency in a 96-well tissue culture plate. Four hours after seeding, 10 µL of titrated VAX014 or VAX-I rBMCs were added at a multiplicity of infection (MOI) concentration range of 2000:1 to 4:1, rBMC-to-mammalian cell ratio at 37 °C, with 5% CO_2_ for 2 or 20 h. Media were replaced prior to measuring cell viability, as described previously [14]. For nuclear staining, cells were incubated with 1.5 μg/mL prior to visualization. In separate experiments, LDH enzyme activity was measured in supernatants after 2 h of co-incubation [14].

### 4.4. Integrin and Propidium Iodide Staining

Integrin and propidium iodide experiments were performed as previously described [14]. Integrin staining was performed using blocking buffer containing 10% heat-inactivated goat serum, 0.2% bovine serum albumin, and 0.02% sodium azide, and 10 µg/mL primary antibodies: anti-human β1 (Cat # 555442, BD Biosciences, San Diego, CA, USA), anti-human α5 (Cat # MAB1956, Millipore, Billerica, MA, USA), anti-human α3 (Cat # sc-13545, Santa Cruz Biotechnology, Santa Cruz, CA, USA) anti-mouse β1 (Cat # 14-0291-81, eBioscience, San Diego, CA, USA), anti-mouse α5 (Cat # 553350, BD Biosciences), anti-mouse α3 (Cat # sc-6588, Santa Cruz Biotechnology) or matching isotype controls (mouse anti-PFO isotype control (Vaxiion Therapeutics, San Diego, CA, USA), hamster isotype control (Cat # 14-4888-81, eBioscience, San Diego, CA, USA) for 30 minutes. Cells were washed and stained with secondary antibodies: goat anti-mouse Alexa Fluor 488, goat anti-hamster Alexa Fluor 488, or donkey anti-goat Alexa Fluor 488 (Life Technologies, Carlsbad, CA, USA) for 30 minutes prior to analysis. Propidium iodide analysis utilized cells treated with VAX014 or VAX-I at a MOI of 2000:1, and were stained propidium iodide after rBMC treatment. Cells analyzed using the Stratedigm S1000 flow cytometer (Stratedigm, San Jose, CA, USA) with a minimum of 50,000 events recorded per sample and analyzed using CellCapture v3.1 software (Stratedigm).

### 4.5. Animal Studies

The Fabp-CreXApc^fl468^ animal model [34] and homozygous Fabp-Cre and Apc^fl468^ parental lines were maintained and housed in a specific pathogen-free vivarium at San Diego State University. Homozygosity of parental lines was verified by PCR. Prior to treatment, animals were sedated using an intraperitoneal injection of ketamine-xylazine cocktail (90 mg/kg ketamine; 10 mg/mL xylazine). VAX014 (1.8 × 10^8^–2.2 × 10^8^ rBMCs), or PBS as a control, was administered intrarectally in volumes sufficient to fill the colon. Animals were monitored daily for any adverse effects resulting from anesthesia or treatment. Euthanasia was carried out using CO_2_ asphyxiation at the end of experimental timepoints or at the onset of severe polyp-related symptoms.

### 4.6. Immunohistochemistry

Colon tissues were Swiss-rolled and formalin-fixed prior to paraffin embedding and sectioning. IHC was performed as previously described [19]. Non-specific protein interactions were blocked with a 1% bovine serum albumin (BSA) for 20 min. Rabbit anti-Ki67 primary antibody (Cat # NB500-170, Novus Biologicals, Littleton, CO, USA) diluted 1:200 in 1% BSA/PBS was incubated for 1 h at room temperature. Antibody binding was detected using EnVision+ (HRP. Rabbit. AEC+) kit (Cat # K4009, Aligent Technologies, Santa Clara, CA, USA), and sections were counter-stained with Mayer’s hematoxylin. α3 (Cat # ab131055, Abcam, Cambridge, UK) and β1 integrin (Cat # ab179471, Abcam, Cambridge, UK) IHC staining was performed using 5 μg/mL primary antibody concentrations and 2 μg/mL biotin-conjugated anti-rabbit secondary antibodies followed by 1 μg/mL streptavidin-HRP reagent to enhance signals. Expression levels of IHC staining in adenomatous polyps were measured using an image analysis protocol by Agley et al. [35], using the Adobe Photoshop imaging software (Version 24.1, CS5 Extended, Adobe Systems). Measurements are reported as a fraction of positive stain to total tumor area in pixels.

### 4.7. Quantitative PCR

RNA was extracted using the Quick-RNA MiniPrep kit (Cat # R1054, Zymo Research, Irvine, CA, USA), normalized, and cDNA was synthesized with the Quanta Q-Script MiniPrep Kit (Cat # 101414-106, Quanta Biosciences, Beverly, MA, USA). Reactions were optimized for a 60 °C annealing temperature and run in duplicate on the CFX96 Touch Real-Time PCR Detection System (Bio-Rad, Hercules, CA, USA) using 25 µL volumes and the PerfeCTa SYBR Green FastMix for IQ (Cat # 101414-254, Quanta Biosciences). Primer sequences are summarized in Appendix A. Gene expression was normalized to a housekeeping gene, either β-actin or RPL32. Statistical analysis was performed using Δ-Ct values of the target and housekeeping genes while results are presented as fold changes normalized to control groups.

### 4.8. Sequencing and Annotation of Colonic Microbiota

Microbiota populations were isolated from tumor adjacent tissues of Fabp-CreXApc^fl468^ animals by mechanical disruption with 0.1 mm Zirconia beads in 0.02 μm filtered SM buffer, as previously described by Reyes et al. [36], and pelleted by centrifugation at 16,000× *g* for 15 min at 4 °C. DNA was extracted using the Nucleospin^®^ Tissue XS DNA extraction Kit (Cat # 740901.50, Machery-Nagel, Bethlehem, PA, USA) and host contamination was minimized with NEBNext Microbiome DNA Enrichment Kit (Cat # E2612L, New England Biosciences, Ipswich, MA, USA). Shotgun metagenomes were generated with the TruSeq Nano DNA Library Preparation kit (Cat # 20015964, Illumina, San Diego, CA, USA), and sequenced using a V3 kit on a MiSeq System (Cat # MS-102-3003, Illumina). Metagenome sequence data are available on NCBI, accession: PRJNA716740.

Sequence quality control was performed using PRINSEQ (Version 020.4) [37] and paired-end reads were joined using PEAR [38]. Host-associated sequence reads were filtered using SMALT (accessed on 18 August 2018) and the C57BL/6 reference genome (https://www.sanger.ac.uk/science/tools/smalt-0). Prokaryotic paired-end reads were annotated for genetic function using SUPER-FOCUS [39] and then assembled into contigs using the SPAdes [40]. Contigs were annotated against the full prokaryotic database on NCBI as of April 2018 using BLASTn and expected value of 1 × 10^−3^ [41]. Species-level diversity analysis was performed using the scikit-bio python package, similar to a method previously reported by Fang et al. [42,43,44].

### 4.9. Statistical Analyses

GraphPad Prism 8 software (GraphPad, La Jolla, CA, USA) was used for statistical analysis and production of figures. Principal component microbiome analysis of weighted UniFrac distance values was assessed for significance using the Analysis of similarities (Anosim) function of the Scikit-bio program package [42]. The non-parametric Mann–Whitney U-test was used for remaining VAX014 analyses.

## Figures and Tables

**Figure 1 ijms-24-09993-f001:**
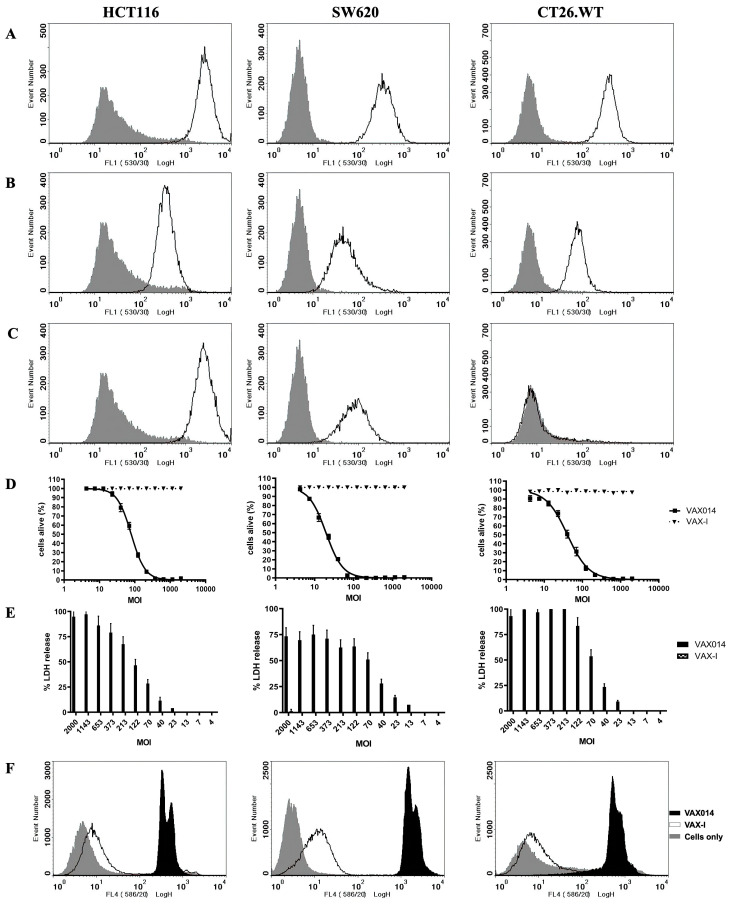
Target validation and VAX014-mediated killing of CRC in vitro. Colon cancer cell lines HCT116, SW620, and CT26.WT were assessed for β1 (**A**) α5 (**B**), and α3 (**C**) integrin expression levels (white with black lines) by FACS analyses when compared to isotype controls (grey). Gates were established using forward and side scatter plots. (**D**) CRC cell viability, measured by Prestoblue™ viability dye, was assessed after 2 h coincubation with increasing concentrations of rBMC to mammalian cell ratios (MOI) of VAX014 and control rBMC VAX-I, lacking the PFO toxin payload. (**E**) Oncolysis via lactate dehydrogenase release (% LDH release) was assessed by lactate dehydrogenase activity in supernatant after 2 h coincubation with VAX014 and VAX-I. (**F**) VAX014-mediated membrane permeabilization was evaluated by FACS analyses of propidium iodide-stained CRC cells treated with equal numbers of VAX-I and VAX014 rBMCs for 2 h and compared to cells only (untreated) control.

**Figure 2 ijms-24-09993-f002:**
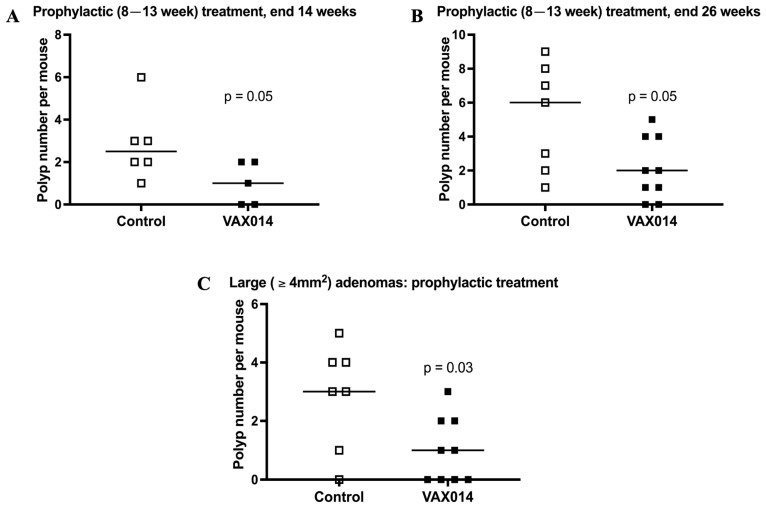
Prophylactic VAX014 adenoma inhibition in the Fabp-CreXApc^fl468^ mouse model of colon cancer. Total polyp number was assessed in Fabp-CreXApc^fl468^ mice treated with VAX014 as a prophylactic (8–13 weeks), administered intrarectally 1X/week, at (**A**) 14 weeks (VAX014 *n* = 6, Control *n* = 5), around the time of macroscopic polyp development, and (**B**) 26 weeks (VAX014 *n* = 9, Control *n* = 7). The abundance of large, ≥4 mm^2^, adenomas (**C**) was quantified between control and VAX014 prophylactic treatment groups (VAX014 *n* = 9, Control *n* = 7).

**Figure 3 ijms-24-09993-f003:**
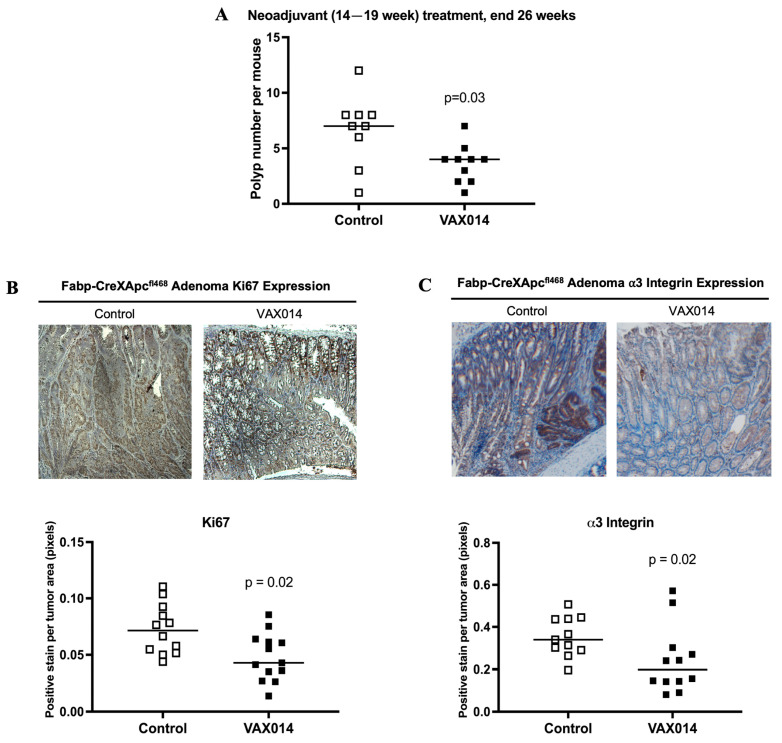
Neoadjuvant VAX014 antitumor activity in vivo. Tumor number was also investigated in neoadjuvant VAX014 mice (treated 14–19 weeks), with weekly intrarectal treatments, at 26 weeks (VAX014 *n* = 10, Control *n* = 9) (**A**). IHC analysis (5X magnification) of the Ki67 proliferation marker (**B**) and α3 integrin (**C**) was assessed in adenomas from Fabp-CreXApc^fl468^ mice treated three times with VAX014 from 22 to 24 weeks of age and harvested 1 h after final treatment. Median values are displayed on graphs. The Mann–Whitney U test was used to evaluate significance.

**Figure 4 ijms-24-09993-f004:**
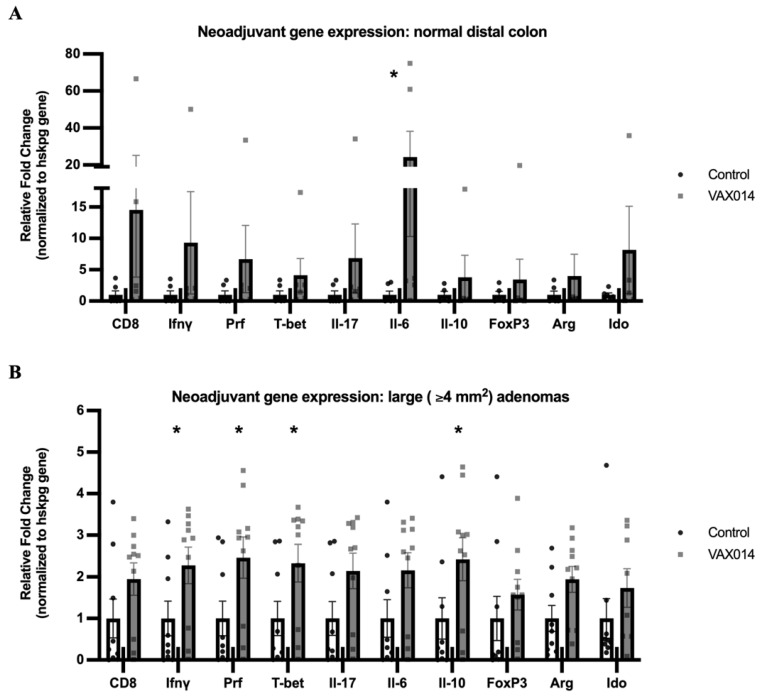
VAX014 induction of antitumor immune gene expression in adenomas. Cytokine markers of inflammation, anti-tumor immune responses, and immune repression were analyzed in colon tissues isolated from Fabp-CreXApc^fl468^ mice treated with neoadjuvant VAX014 (14–19 weeks). Analysis of immune gene markers isolated from macroscopically normal colon tissues (**A**) (VAX014 *n* = 6, Control *n* = 6) and large, ≥4 mm^2^, adenomas (**B**) (VAX014 *n* = 10, Control *n* = 9) harvested and frozen at 26 weeks. Means with standard error of the mean (SEM) are displayed on graphs; the Mann–Whitney U test was used to test significance. * *p* < 0.05.

**Figure 5 ijms-24-09993-f005:**
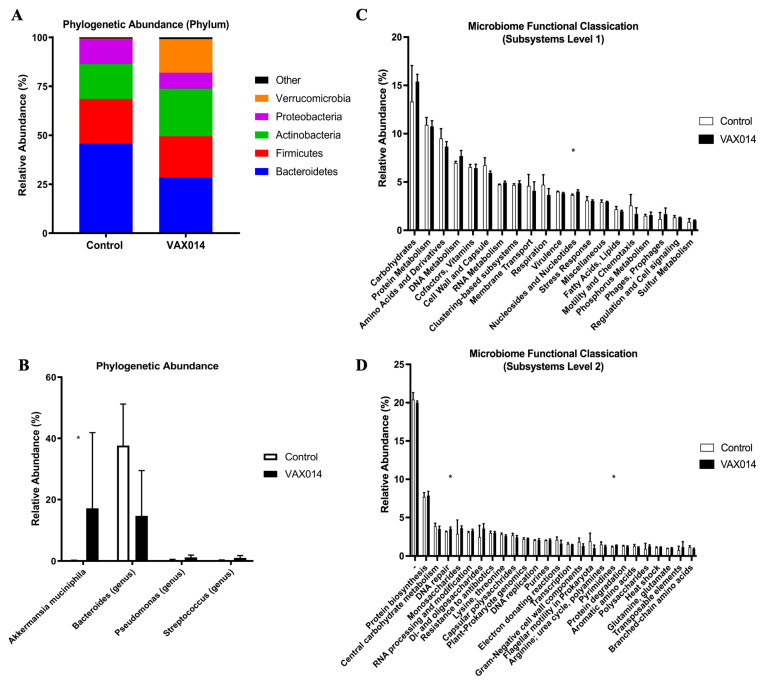
VAX014 induced modulation of the colonic microbiome. Microbiomes isolated from colon tissues directly adjacent to adenomatous polyps from 26-week-old mice from the neoadjuvant treatment group (VAX014 *n* = 5, Control *n* = 3). Taxonomic abundance of microbiome populations was assessed at the phylum (**A**) and species (**B**) levels of classification using assembled contigs and all NCBI sequenced prokaryotes, as of April 2018. Annotation of gene function using paired-end reads, the SEED systems database, and SUPER-FOCUS annotation evaluated gene abundances in colonic microbiome at Subsystems level 1 (**C**) and Subsystems level 2 (**D**) categories. Mean values with standard deviations are represented on graphs. Statistical significance was assessed using the Mann–Whitney U test. * *p* < 0.05.

## Data Availability

Immunohistochemistry, Fabp-CreXApc^fl468^ metadata, and raw qRT-PCR data can be found at: https://figshare.com/projects/VAX014_an_oncolytic_therapy_reduces_adenomas_and_modifies_colon_microenvironment_in_mouse_model_of_CRC/100820. Metagenomic sequence data are available on NCBI, accession #PRJNA716740.

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
