# Peer review of "VAX014, an Oncolytic Therapy, Reduces Adenomas and Modifies Colon Microenvironment in Mouse Model of CRC"

_ijms, 2023, doi:10.3390/ijms24129993_

Round 1
Reviewer 1 Report
In the proposed manuscript, Grenier and Colleagues analyzed the effect of VAX014 on colorectal cancer. VAX014 is an oncolytic bacterial minicell-based therapy that showed an antitumor effect on bladder cancer, but not yet on CRC. The experiments described by the Authors are correctly designed and quite convincing about the efficacy of the proposed treatment; however, the topic does not exactly represent a novelty in the field considering that VAX014 is currently at the clinical stage, even if in a different disease setting.
The following major issues should be addressed.
Ref19 proves that VAX-I does not induce antitumor activity in bladder cancer models, and for this reason this control was not included in the proposed paper. However, the tumor model of the present study (colon) significantly differs from the one presented in ref19 (bladder). To this reviewer it would be important to demonstrate the lack of VAX-I activity also on CRC, at least as a proof-of-concept.
Fig4 shows cytokine markers of inflammation, anti-tumor immune responses and immune repression. PCR performed on frozen tissues is not the ideal assay for this kind of measurements, and important outliers among the fold change levels can be observed. On the contrary, immunoassays able to capture the secreted molecules are more reliable in cytokines measurement and, similarly, flow cytometry analyses of intratumoral immune cell populations can better characterize the immune phenotype induced by the treatment.
Microbiota analyses were performed on colonic tissue associated to the tumor. However, data in literature are not unanimous in considering colonic adjacent tissue a good representation of colonic tumor tissue in terms of bacterial population; other publications also report the analysis of microbiota performed on fecal samples.
Metabolomics and 16S rRNA sequencing of human colorectal cancers and adjacent mucosa (https://doi.org/10.1371/journal.pone.0208584)
Comparison of Gut Microbiome in Human Colorectal Cancer in Paired Tumor and Adjacent Normal Tissues (https://doi.org/10.2147/OTT.S218004)
Intestinal bacteria detected in cancer and adjacent tissue from patients with colorectal cancer (https://doi.org/10.3892/ol.2018.9714)
Microbiota analyses should be extended to tumor tissues and / or fecal samples.
Target validation and VAX014-mediated killing should be tested also ex vivo, similarly to in vitro experiments performed on cell lines.
349-350: The statement “a lack of β1 integrin reduction by neoadjuvant VAX014 may be due to variable expression observed in vivo” is not clear to this reviewer. Considering that VAX014 should interact with both a3 and β1, how could the Authors explain the different modulation of the two integrin isoforms?
It would be interesting to evaluate, for each mouse, the integrin levels together with the polyp number shown in Fig3A, in order to verify if there is a correlation that could explain the observed heterogeneity.
CT26 cell line, used for in vitro preliminary experiments, could be useful also in an in vivo setting. BALB/c syngeneic mice can be used to inject cells, eg orthotopically, obtaining a tumor model that is more “artificial” compared to the one exploited in the manuscript, but at the same time more reproducible in terms of tumor volumes and growth rates. The advantage of the proposed model is based on its spontaneity indeed, but this translates in a low or null number of tumor lesions in the control groups, that could impact on data interpretation.
Minor points:
130: lacking space
174: double space
180: double space
188: hours
Fig4B: adenomas
387: double space
Quality of fig 1 is poor
Author Response
Responses to Reviewer 1 Comments
Point 1: In the proposed manuscript, Grenier and Colleagues analyzed the effect of VAX014 on colorectal cancer. VAX014 is an oncolytic bacterial minicell-based therapy that showed an antitumor effect on bladder cancer, but not yet on CRC. The experiments described by the Authors are correctly designed and quite convincing about the efficacy of the proposed treatment; however, the topic does not exactly represent a novelty in the field considering that VAX014 is currently at the clinical stage, even if in a different disease setting. The following major issues should be addressed.
Response 1: We appreciate the feedback regarding our evaluation of VAX014 in CRC, we however feel that the novelty of this manuscript is due to use of the orthologous mouse model, for which VAX014 has not been previously evaluated, and the inclusion of the colonic microbiome, which has not been evaluated with VAX014 prior to this experimentation. Previously, VAX014 has been evaluated in orthotopic MB49 mouse model (10.1038/mto.2016.4., 10.21873/anticanres.11218, and 10.1158/2326-6066.CIR-21-0879) but never in a genetically compromised model such as the Fabp-CreXApcfl468 preclinical model which may more accurately recapitulate the immune environment of CRC patients. We feel that the combination of both the use of this animal model and inclusion of the colonic microbiome represent a marketed advancement of VAX014 research which may prove relevant for future clinical testing.
Point 2: Ref19 proves that VAX-I does not induce antitumor activity in bladder cancer models, and for this reason this control was not included in the proposed paper. However, the tumor model of the present study (colon) significantly differs from the one presented in ref19 (bladder). To this reviewer it would be important to demonstrate the lack of VAX-I activity also on CRC, at least as a proof-of-concept.
Response 2: While we appreciate that VAX-I could be used as a negative control for these experiments, we did not use it for two main reasons. One is that in vitro studies have shown that VAX-I is no more active in colon cancer cell lines (Figure 1) than in any other type of cancer, including bladder cancer cell lines and B16F10, a murine malignant melanoma. The orthotopic bladder cancer model studies, both in vitro and in vivo, have been published (10.1038/mto.2016.4., 10.21873/anticanres.11218, and 10.1158/2326-6066.CIR-21-0879) and the studies on B16F10 are currently in press with the Journal for Immunotherapy of Cancer (Reil, K. et al., 2023. Intralesional administration of VAX014 facilitates in situ immunization and potentiates immune checkpoint blockade in immunologically cold tumors. JITC In Press - we are reviewing galley proofs now). All these studies have shown that VAX-I has no activity, no matter the cancer model. We would be happy to provide a draft copy of this in press manuscript if it is necessary for the evaluation of our work in the current manuscript.
The second, most compelling, reason we have used saline as a control is that every publication using VAX014 in in vivo studies so far, including all those in orthotopic bladder cancer models (10.1038/mto.2016.4., 10.21873/anticanres.11218, and 10.1158/2326-6066.CIR-21-0879), as well as the new B16F10 manuscript currently in press (Reil, K et al., 2023, Journal for Immunotherapy of Cancer) have used saline as a control and we want our studies to be directly comparable to those studies. Because IACUC standards limit the number of animals used in our experimentation to those that are absolutely necessary, we made the judgement call to eliminate the additional negative control of VAX-I for our in vivo studies. Thus, the VAX-I controls are not available for this work. We respectfully suggest that due to the work that has come before, saline is the appropriate negative control for these studies.
Point 3: Fig4 shows cytokine markers of inflammation, anti-tumor immune responses and immune repression. PCR performed on frozen tissues is not the ideal assay for this kind of measurements, and important outliers among the fold change levels can be observed. On the contrary, immunoassays able to capture the secreted molecules are more reliable in cytokines measurement and, similarly, flow cytometry analyses of intratumoral immune cell populations can better characterize the immune phenotype induced by the treatment.
Response 3: While we agree that flow cytometry or other immunological techniques would be more accurate in describing specific anti-tumor immune responses were that to be the goal of this manuscript, we wanted to do analyses on individual tumors, not pooled samples. Therefore, the qPCR experiments used in this manuscript were designed to provide a broad evaluation of the immune microenvironment in VAX014 treated Fabp-CreXApcfl468 polyps. Because our aim was to provide an overview of immune gene expression in individual tumors, this would have not been possible using either flow cytometry or ELISA where single adenomas (ranging from 0.5-9mm2) or colon tissues would not meet the input requirements for these assays. Because we never intended to provide a complete and thorough analysis of immunity in the tissues of these mice, we acknowledge these immunoassay limitations and we have now emphasized the limitations of our experiments on page 11, lines 378-381.
Point 4: Microbiota analyses were performed on colonic tissue associated to the tumor. However, data in literature are not unanimous in considering colonic adjacent tissue a good representation of colonic tumor tissue in terms of bacterial population; other publications also report the analysis of microbiota performed on fecal samples.Metabolomics and 16S rRNA sequencing of human colorectal cancers and adjacent mucosa (https://doi.org/10.1371/journal.pone.0208584) Comparison of Gut Microbiome in Human Colorectal Cancer in Paired Tumor and Adjacent Normal Tissues (https://doi.org/10.2147/OTT.S218004)
Intestinal bacteria detected in cancer and adjacent tissue from patients with colorectal cancer (https://doi.org/10.3892/ol.2018.9714) Microbiota analyses should be extended to tumor tissues and / or fecal samples.
Response 4: We acknowledge that microbiome literature is not unanimous in supporting tumor adjacent tissue sampling as a representative CRC-associated microbiota. However, both tumor tissue and fecal microbiota sampling methods have been demonstrated to possess similar disadvantages and biases in representing the colonic microbiota. A review of different intestinal microbiota sampling methods (https://doi.org/10.3389/fcimb.2020.00151) describes the inaccuracy of determining changes in gut microbiota from fecal samples and the insufficiency of biomass yields from tumor samples, while identifying distinct mucosa and fecal microbiota populations. Further, fecal microbiome sampling is also subject to temporal and subsampling variations (10.1038/s41598-021-93031-z) while tumor microbiota populations may change based on adenoma location (https://doi.org/10.3389/fmicb.2021.727937). For all of these reasons, we felt that tumor adjacent tissue had the ability to provide insights, even if the complete story remains unclear.
However, it should be noted that in a study by Wang et al. 2022 (https://doi.org/10.1016/j.celrep.2022.111890) fecal transplantation of the the bacterium Akkermansia muciniphilia significantly suppressed the development of Ovarian cancer through the stimulation of CD8+ T cells. Further, Akkermansia muciniphilia abundabce was associated with murine intestinal diseases using both fecal and luminal samples (10.1038/ismej.2012.39). Therefore, it seems highly probable that our observation is accurate independent of sampling method.
To clarify the relevancy Akkermansia muciniphilia in our study, the references described above have been added to our manuscript (Page11, Lines 400-402 and 404).
Point 5: Target validation and VAX014-mediated killing should be tested also ex vivo, similarly to in vitro experiments performed on cell lines.
Response 5: We agree that this would be a powerful technique for future studies. However, given that ex vivo techniques have not been developed for the Fabp-CreXApcfl468.model or for the evaluation of VAX014 in any preclinical publications (see references above), we believe that these experiments are outside the scope of this proof-of-concept study.
Point 6: 349-350: The statement “a lack of β1 integrin reduction by neoadjuvant VAX014 may be due to variable expression observed in vivo” is not clear to this reviewer. Considering that VAX014 should interact with both a3 and β1, how could the Authors explain the different modulation of the two integrin isoforms? It would be interesting to evaluate, for each mouse, the integrin levels together with the polyp number shown in Fig3A, in order to verify if there is a correlation that could explain the observed heterogeneity.
Response 6: We agree that this statement needed clarification and has been updated (page 11 line 361-362). During these in vivo studies we observed that levels of β1 integrin expression varied by adenoma in a single animal within the control group. While, VAX014 has been demonstrated to target both a3b1 and a5b1, β1 integrins can dimerize with 12 distinct a subunits (10.1038/s41390-020-01177-9.) as mentioned in the discussion but not all integrin dimers may be targeted by VAX014, leading us to propose that the heterogeneity of these dimers may also contribute to the lack of observed differences. This phrase regarding integrin dimers (page 11, line 363) has been edited for clarity.
While it would have been interesting to compare integrin levels from tumor numbers, β1 integrin levels were measured from a different treatment group than the polyp number shown in Fig3A. In figure 3A, Fabp-CreXApcfl468 mice were treated in the neoadjuvant group, from 14 to 19 weeks with either VAX014 or controls, and sacrificed at 26 weeks of age, as described in the text. Since tumor loads were found to be significantly reduced in this group, an additional treatment group, treated from 22-24 weeks of age and sacrificed 1 hour after the third dose, were included to investigate the local effects of VAX014 treatment in the presence of well-developed adenomas where Ki67, a3, a5, and β1 integrins were evaluated by immunohistochemistry. Because of this design, tumor load was not expected to differ between VAX014 and control treatment groups for the adenomas in which integrins were evaluated.
Point 7: CT26 cell line, used for in vitro preliminary experiments, could be useful also in an in vivo setting. BALB/c syngeneic mice can be used to inject cells, eg orthotopically, obtaining a tumor model that is more “artificial” compared to the one exploited in the manuscript, but at the same time more reproducible in terms of tumor volumes and growth rates. The advantage of the proposed model is based on its spontaneity indeed, but this translates in a low or null number of tumor lesions in the control groups, that could impact on data interpretation.
Response 7: We agree that the use of the CT26 orthotopic model is a powerful model to be used in the evaluation of VAX014. In the manuscript currently in press with the Journal for Immunotherapy of Cancer (Reil, K., et al., 2023) CT26 was used as a secondary model for the evaluation of VAX014 efficacy against various tumor types (we would be happy to provide a draft copy of this in press manuscript if it is necessary for the evaluation of our work in the current manuscript). Using the orthologous CT26 model, we were able to show that VAX014 treatment decreased tumor burden (p=0.01) and increased survival (p=0.001) for mice bearing these tumors. 7/20 mice had a complete rejection of their CT26 tumor after VAX014 intratumoral treatment (6 doses, 1 week apart) while 0/5 saline controls rejected their tumors. The response was found to be completely dependent on CD8+ T cells and in mice that rejected their tumors, 6/7 rejected CT26 when rechallenged with untreated tumors several weeks later. These data establish that VAX014 induces long term immunological memory against CT26 in some mice. So, yes, we agree with the value of this model. However, this type of study was not the goal of the current manuscript under review.
For the study under review here, we deliberately chose to utilize Fabp-CreXApcfl468 to more accurately recapitulate the natural immunological and microbiome microenvironment in the development of adenomas. Since VAX014 is of bacterial origin, the effects of administration into the local colonic environment were a major consideration in safety and efficacy of this therapeutic, and the logical basis for using this model. The potential for low tumor lesions in the model was also considered when using the CreXApcfl468 model; to compensate experimental populations were increased, producing a total of 183 adenomas which were evaluated for VAX014 efficacy across treatment groups. Therefore, we respectfully suggest that sample numbers were not limited in our study.
Because the CT26 data is now in press we have made minor modifications to the abstract, to be sure that it is accurate with respect to the efficacy of VAX014 in CRC. (Page 1, Line 12). The abbreviation to TH1 was also removed from the abstract to meet abstract word-limit requirements (Page 1, Line 17). Since this manuscript was not the first to evaluate VAX014 in the context of CRC, (Page 2, Line 83) was modified to clarify the context of this study.
Minor points:
130: lacking space (Fixed)
174: double space (Fixed)
180: double space (Fixed)
188: hours
Response: We respectfully decided not to make this correction because the pleural is already indicated in the word ‘coincubations’ that follows the two time points.
Fig4B: adenomas (Fixed)
387: double space (Fixed)
Quality of fig 1 is poor
Response: The formatting and legends of figure 1 pictures have been edited to improve quality and a new figure one has been inserted into the manuscript (page 5)
Reviewer 2 Report
The authors have explored the use of an oncolytic bacteria minicell-based therapy (VAX014) in a prophylactic setting for inhibition of adenoma development. It is indeed an effective one and it modifies colon microenvironment in a mouse model of CRC. Data are mostly solid.
There are some minor issues.
1. Figure 1A. Authors have checked beta1, alpha5 and alpha3. Why beta5 not included as it is also important (page 2, line 75)?
2. Figure 4A, 4B. It is interesting to find that all biomarkers examined showed a trend of elevation in VAX014-treated when compared to the control. These data are expressed relative to a house-keeping gene (either beta-actin or RPL32). As we know, some house-keeping genes are heavily regulated by treatments such as chemotherapy and immunotherapy. Have you checked this possibility and are results the same (or similar) if either control is used?
3. Please check some sentences. For example, page 3 line 146. The phrase “reverse transcribed with” should be “reversely transcribed with”.
4. It is weird to see that many references are labelled with (in Eng). I thought that most, if not 100%, references are in English. If one is not in English, then it is more likely to be labelled with (not in Eng) or whatever.
5. Reference #1. If you go to the American Cancer Society page, they tell you that these facts and figures are a part of formal publication. Thus, it is better to cite that publication:
6. Siegel RL, Wagle NS, Cercek A, Smith RA, Jemal A. Colorectal cancer statistics, 2023. CA Cancer J Clin. 2023 May-Jun;73(3):233-254. [PMID: 36856579]
7. Missing information in References:
Ref #8. Article number (AN)?
Ref #31. AN?
Ref #33. AN?
Minor
Round 2
Reviewer 1 Report
Authors addressed the issues proposed by the reviewer and supplied the additional information required to support their data. The updated version of the manuscript is acceptable for publication to this reviewer.